# Statistical Postprocessing of Ensemble Forecasts for Severe Weather at Deutscher Wetterdienst

Reinhold Hess

Deutscher Wetterdienst

**Correspondence:** Reinhold Hess (reinhold.hess@dwd.de)

**Abstract.** This paper gives an overview of DWD's postprocessing system called Ensemble-MOS together with its motivation and the design consequences for probabilistic forecasts of extreme events based on ensemble data. Forecasts of the ensemble systems COSMO-D2-EPS and ECMWF-ENS are statistically optimised and calibrated by Ensemble-MOS with a focus on severe weather in order to support the warning decision management at Deutscher Wetterdienst (DWD).

Ensemble mean and spread are used as predictors for linear and logistic multiple regressions to correct for conditional biases. The predictands are derived from synoptic observations and include temperature, precipitation amounts, wind gusts and many more, and are statistically estimated in a comprehensive model output statistics (MOS) approach. Long time series and collections of stations are used as training data that capture a sufficient number of observed events, as required for robust statistical modelling.

Logistic regressions are applied to probabilities that predefined meteorological events occur. Details of the implementation including the selection of predictors with testing for significance are presented. For probabilities of severe wind gusts global logistic parameterisations are developed that depend on local estimations of wind speed. In this way, robust probability forecasts for extreme events are obtained while local characteristics are preserved.

The problems of Ensemble-MOS, such as model changes and consistency requirements, which occur with the operative
MOS systems of the DWD are addressed.

# 1 Introduction

Ensemble forecasting rose with the understanding of the limited predictability of weather. This limitation is caused by sparse and imperfect observations, approximating numerical data assimilation and modelling and by the chaotic physical nature of the atmosphere. The basic idea of ensemble forecasting is to vary observations, initial and boundary conditions, and physical parameterisations within their assumed scale of uncertainty, and rerun the forecast model with these changes.

The obtained ensemble of forecasts expresses the distribution of possible weather scenarios to be expected. Probabilistic forecasts can be derived from the ensemble, like forecast errors, probabilities for special weather events, quantiles of the distribution or even estimations of the full distribution. The ensemble spread is often used as estimation for forecast errors. In a perfect ensemble system the spread is statistically consistent with the forecast error of the ensemble mean against observations (Wilks, 2011, e.g.), however, it is experienced often too small, especially for near surface weather elements and short lead times. Typically, an optimal spread-skill relationship close to 1 and its involved forecast reliability is much easier obtained for atmospheric variables in higher vertical layers, as e.g. $500\,\mathrm{hPa}$ geopotential height, than for screen level variables like $2\,\mathrm{m}$ temperature, $10\,\mathrm{m}$ wind speed or precipitation (Buizza et al., 2005; Gebhardt et al., 2011; Buizza, 2018, e.g.); see also Sec. 4.

In order to make best use of the probabilistic information contained in the ensembles, e.g. by relating probabilities for harmful weather events with economical value in cost-loss evaluations (Wilks, 2001; Ben Bouallègue et al., 2015, e.g.), the ensemble forecasts should be calibrated to observed relative frequencies as motivated by Buizza (2018). Warning thresholds are the levels of probabilities at which meteorological warnings are to be issued. These thresholds may be tailored to the public depending on categorical scores such as probability of detection (POD) and false alarm ratio (FAR). Statistical reliability of forecast probabilities is considered essential for qualified threshold definitions and for automated warning guidances.

For deterministic forecasts statistical postprocessing is used for optimisation and interpretation. This is likewise true for ensemble forecasts, where statistical calibration is an additional application of postprocessing. Gneiting et al. (2007) describes probabilistic postprocessing as a method to maximise the sharpness of a predictive distribution under the condition of calibration (the climatologic average is calibrated too, however it has no sharpness and is useless as a forecast). Nevertheless, optimisation is still an issue for ensemble forecasts. In general, the systematic errors of the underlying numerical model turn up in each forecast member and thus are retained in the ensemble mean. Averaging only reduces the random errors of the ensemble members.

Due to its ability to improve skill and reliability of probabilistic forecasts, many different postprocessing methods exist for both single- and multi-model ensembles. There are comprehensive multivariate systems and univariate systems, that are specific to a certain forecast element. Length of training data generally depends on the statistical method and application, however, the availability of data is also often a serious limitation. Some systems perform individual training for different locations in order to account for local characteristics, whilst others apply the same statistical model to collections of stations or grid points. Global modelling improves statistical sampling at the cost of orographic and climatologic disparities.

Classical MOS systems tend to underestimate forecast errors if corrections are applied to each ensemble member individually. In order to maintain forecast variability, Vannitsem (2009) suggests considering observation errors. Gneiting et al. (2005)

proposes non-homogeneous Gaussian regression (NGR) that relies on Gaussian distributions. The location and scale parameters of the Gaussian distributions correspond to a linear function of the ensemble mean and ensemble spread, respectively. The NGR coefficients are trained by minimising the continuous ranked probability score (CRPS). In Bayesian model averaging (BMA) (Raftery et al., 2005; Möller et al., 2013, e.g.) distributions of already bias corrected forecasts are combined as weighted averages using kernel functions.

Many different postprocessing methods tailored to different variables exist, only some are mentioned here. For 24-hourly precipitation Hamill (2012) presents a multimodel ensemble postprocessing based on extended logistic regression and eight years of training data. Hamill et al. (2017) describe a method to blend high-resolution multimodel ensembles by quantile mapping with short training periods of about two months for 6 and 12-hourly precipitations. Postprocessing methods specialising in wind speed have been developed as well, e.g. Sloughter et al. (2013) uses BMA in combination with Gamma distributions. An overview of conventional univariate postprocessing approaches is given in Wilks (2018).

In addition to the univariate postprocessing methods mentioned above, there exist also approaches to model spatio-temporal dependence structures, and hence, to produce ensembles of forecast scenarios. This enables, for instance, to estimate area related probabilities. Schefzik et al. (2013); Schefzik and Möller (2018) use ensemble copula coupling (ECC) and Schaake shuffle-based approaches in order to generate postprocessed forecast scenarios for temperature, precipitation and wind. Ensembles of ECC forecast scenarios provide high flexibility in product generation to the constraint that all ensemble data is accessible.

Fewer methods focus on extreme events of precipitation and wind gusts that are essential for automated warning support. Friederichs et al. (2018) use the tails of generalised extreme-value distributions in order to estimate conditional probabilities of extreme events. As extreme meteorological events are (fortunately) rare, long time series are required to capture a sufficiently large number of occurred events in order to derive statistically significant estimations. For example, strong precipitation events with rain amounts of more than $15\,\mathrm{mm}$ per hour are captured only about once a year at each rain gauge within Germany. Extreme events with more than $40\,\mathrm{mm}$ and $50\,\mathrm{mm}$ rarely appear, nevertheless warnings are essential when they do.

With long time series, a significant portion of the data consists of calm weather without relevance for warnings. It is problematic, however, to restrict or focus training data on severe events. In doing so, predictors might be selected that are highly correlated to the selected series of severe events, but accidentally also to calm scenarios that are not contained in the training data. In order to exclude these spurious predictors and to derive skilful statistical models, more general training data need to be used, since otherwise overforecasting presumably results and frequency bias (FB) and FAR increase. This basically corresponds to the idea of the *forecaster's dilemma*, see Lerch et al. (2017), that states that overforecasting is a promising strategy when forecasts are evaluated mainly for extreme events.

The usage of probabilistic forecasts for warnings of severe weather also influences the way the forecasts need to be evaluated. Also for verification, long time periods are required to capture enough extreme and rare events to derive statistically significant results. Verification scores like root mean square error (RMSE) or CRPS (Hersbach, 2000; Gneiting et al., 2005, e.g.) are highly dominated by the overwhelming majority of cases when no event occurred. Excellent but irrelevant forecasts of calm weather can pretend good verification results, although the few relevant extreme cases might not be forecasted well. Categorical scores

like POD and FAR are considered more relevant for rare and extreme cases, along with other more complex scores like Heidke Skill Score (HSS) or equitable threat score (ETS). Also scatter diagrams reveal outliers and are sensitive to extreme values.

Here we present a MOS approach that has been tailored to postprocessing ensemble forecasts for extreme and rare events. It is named Ensemble-MOS and has been set up at DWD in order to support warning management with probabilistic forecasts of potentially harmful weather events within AutoWARN, see Reichert et al. (2015); Reichert (2016, 2017). Altogether 37 different warning elements exist at DWD, including heavy rain and strong wind gusts, both at several levels of intensity, thunderstorms, snowfall, fog, limited visibility, frost and others. Currently the ensemble systems COSMO-D2-EPS and ECMWF-ENS are statistically optimised and calibrated using several years of training data, but Ensemble-MOS is applicable to other ensembles in general.

At DWD, statistically postprocessed forecasts of the ensemble systems COSMO-D2-EPS and ECMWF-ENS and also of the deterministic models ICON and ECMWF-IFS are combined in a second step in order to provide a consistent data set and a seamless transition from very short-term to medium range forecasts. This combined product provides a single voice basis for the generation of warning proposals, see Reichert et al. (2015). The combination is based on a second MOS approach similar to the system described here; it uses the individual statistical forecasts of the numerical models as predictors. As a linear combination of calibrated forecasts not necessarily preserves calibration (Ranjan and Gneiting, 2010, e.g.), additional constant predictors are added to the MOS-equations as a remedy. Primo (2016); Reichert (2017) state that automated warnings of wind gusts based on the combined product achieve a performance that is comparable to that of human forecasters.

The further outline of the paper is as follows: After the introduction, the used observations and ensemble systems are introduced in Sec. 2. Thereafter, Sec. 3 describes the conceptual design of Ensemble-MOS with the definition of predictands and predictors and provides technical details of the stepwise linear and logistic regressions. Especially for extreme wind gusts, a global logistic regression is presented that uses statistical forecasts of the speed of wind gusts as predictors for probabilities of strong events. General caveats of MOS like model changes and forecast consistency are addressed at the end of that section. The results shown here focus on wind gusts and are provided in Sec. 4. Finally, Sec. 5 provides a summary and conclusions.

## 2 Observations and ensemble data

Synoptic observations and model data from the ensemble systems COSMO-D2-EPS and ECMWF-ENS are used as training data and for current statistical forecasts. Time series of eight years of observations and model data have been gathered for training at the time of writing. The used data are introduced in the following.

### 2.1 Synoptic observations

Observations of more than 320 synoptic stations within Germany and its surroundings are used as part of the training data. For short time forecasts the latest available observations at run time are used also as predictors for the statistical modelling, which is described in more detail in Sec. 3.1.

The synoptic observations include measurements of temperature, dew point, precipitation amounts, wind speed and direction, speed of wind gusts, surface pressure, global radiation, visibility, cloud coverage at several height levels, past and present weather and many more. Past and present weather also contain observations of thunderstorm, kind of precipitation and fog amongst others. Ensemble-MOS derives all predictands that are relevant for weather warnings based on synoptic measurements in a comprehensive approach in order to provide the corresponding statistical forecasts. In this paper, we focus on the speeds of wind gusts and on probabilities for severe storms.

## 2.2 COSMO-D2-EPS and upscaled precipitation probabilities

The ensemble system COSMO-D2-EPS of DWD consists of 20 members of the numerical model COSMO-D2. It provides short-term weather forecasts for Germany with runs every three hours (i.e. 00 UTC, . . . , 21 UTC) with forecast steps of 1 h up to 27 h ahead (up to 45 h for 03 UTC). COSMO-D2 was upgraded from its predecessor model COSMO-DE on 15 May 2018, together with its ensemble system COSMO-D2-EPS; the upgrade included an increase in horizontal resolution from 2.8 km to 2.2 km and an adapted orography. Detailed descriptions of COSMO-DE and its ensemble system COSMO-DE-EPS are provided in Baldauf et al. (2011) and Gebhardt et al. (2011); Peralta et al. (2012), respectively. For the ensemble system initial and boundary conditions as well as physical parameterisations are varied according to their assumed levels of uncertainty.

For the postprocessing of COSMO-D2-EPS, eight years of data have been gathered including data from the predecessor system COSMO-DE-EPS, which has been available since 8 December 2010. Thus, a number of model changes and updates are included in the data; impacts on statistical forecasting are addressed later in Sec. 3.4. Each run of Ensemble-MOS starts two hours after the corresponding run of COSMO-D2-EPS to assure that the ensemble system has finished and the data are available.

Forecast probabilities of meteorological events can be estimated as the relative frequency of the ensemble members that show the event of interest. If the relative frequencies are evaluated grid point by grid point, the probabilities imply that the event occurs within areas of the sizes of the grid cells, which are $2.2 \times 2.2 \, \text{km}^2$ for COSMO-D2-EPS. It is therefore not straightforwardly possible to compare event probabilities of ensembles of numerical models with different grid resolutions.

For near surface elements and short lead times the COSMO-D2-EPS is often underdispersive and underestimates forecast errors. Figure 1 shows a rank histogram for 1-hourly precipitation amounts of COSMO-DE-EPS. Too many observations have either less or more precipitation than all members of the ensemble. Using these relative frequencies as estimations of event probabilities statistically results in too many probabilities with values 0 and 1.

Because of the high spatial variability of precipitation, also upscaled precipitation products are derived from COSMO-D2-EPS, which are relative frequencies of precipitation events within areas of $10 \times 10$ grid points (i.e. $22 \times 22 \, \text{km}^2$). A meteorological event (e.g. that the precipitation rate exceeds a certain threshold) is considered to occur within an area, if the event occurs at at least one of its grid points. Area probabilities are therefore estimated straightforwardly as the relative number of ensemble members predicting the area event, not requiring that the event takes place at exactly the same grid point for all ensemble members.

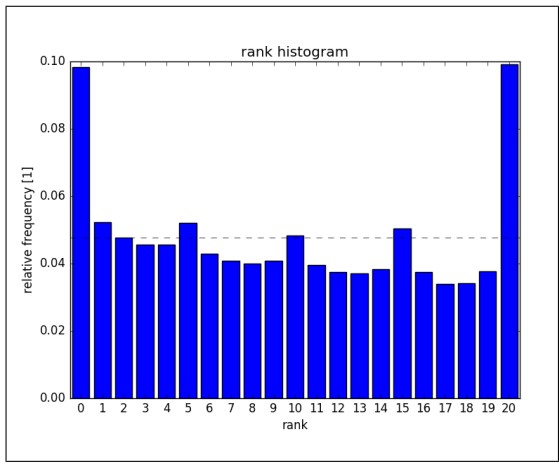

**Figure 1.** Rank/Talagrand histogram for 1-hourly precipitation amounts of COSMO-DE-EPS, forecast lead time 3 h, data for 18 stations from 2011 to 2017

Certainly, also these raw ensemble based estimates are affected by systematic errors of the numerical model COSMO-D2. Hess et al. (2018) observed a bias of -6.2 percentage points for the upscaled precipitation product of COSMO-DE-EPS for the probability that hourly precipitation rate exceeds 0.1 mm. Verification has been done against gauge adjusted radar observations, which is a suitable observation system for areas.

## 2.3 ECMWF-ENS and TIGGE-data

The ECMWF-ENS is a global ensemble system based on the Integrated Forecasting System (IFS) of the European Centre for Medium-Range Weather Forecasts (ECMWF). It consists of 50 perturbed members plus one control run and is computed twice a day for 00 UTC and 12 UTC up to 15 days ahead (and even further with reduced resolution). Postprocessing of Ensemble-MOS at DWD is based on the 00 UTC-run with forecast lead times up to 10 days in steps of three hours. Forecasts of ECMWF-ENS are interpolated from their genuine spectral resolution to a regular grid with 28 km (0.25°) mesh size. Data have been gathered according to the availability of COSMO-DE/2-EPS since 8 December 2010.

TIGGE-data from 2002 to 2013, see Bougeault et al. (2010); Swinbank et al. (2016) of ECMWF-ENS were used in a study to demonstrate the benefits of Ensemble-MOS prior to unarchiving and downloading the gridded ensemble data mentioned above. This study was restricted to the available set of model variables of TIGGE (2 m temperature, mean wind, cloud coverage and 24 h precipitation); results are given in Sec. 4.2.

## 3 Postprocessing by Ensemble-MOS

The Ensemble-MOS of DWD is a model output statistics (MOS) system specialised to postprocess the probabilistic information of NWP-ensembles. Besides calibrating probabilistic forecasts, Ensemble-MOS simultaneously optimises also continuous

variables, as e.g. precipitation amounts and the speeds of wind gusts. Moreover, also statistical interpretations exist for meteorological elements that are not available from numerical forecasts (e.g. thunderstorm, fog or range of visibility). In principle, all meteorological parameters and events that are regularly observed can be forecasted statistically. This includes temperature, dew point, wind speed and direction, wind gusts, surface pressure, global radiation, visibility, cloud coverage at several height levels as well as the synoptic weather with events of thunderstorm, special kinds of precipitation, fog and more.

The basic concept of Ensemble-MOS is to use ensemble mean, spread, and other ensemble statistics as predictors in multiple linear and logistic regressions. The use of ensemble products as predictors instead of processing each ensemble member individually prevents difficulties with underdispersive statistical results and underestimated errors especially for longer forecast horizons. Since MOS systems usually tend to converge towards climatology due to the fading accuracy of numerical models and the limited predictability of meteorological events (Vannitsem, 2009, e.g.), individually processed members converge accordingly. Moreover, multivariate MOS systems perform corrections that depend on the selected set of predictors in order to reduce conditional biases. If the postprocessing of the individual ensemble members uses the same set of predictors, the resulting statistical forecasts become correlated and underdispersive also for this reason.

Ensemble-MOS is based on a MOS system originally set up for postprocessing deterministic forecasts of the former global numerical model GME of DWD and of the deterministic, high resolution IFS of ECMWF, see Knüpffer (1996). Using the ensemble mean and spread as model predictors allows applying the original MOS-approach for deterministic NWP-models to ensembles in a straightforward way.

For continuous variables, such as temperature, precipitation amount or wind speed, deterministic Ensemble-MOS forecasts and estimates of the associated forecast errors are obtained by multiple linear regression. The MOS-equation of a statistical estimate $\hat{y}_k$ with $k$ predictors $x_1, \ldots, x_k$ and $k+1$ coefficients $c_0, \ldots, c_k$ of a continuous predictand $y$ is

$$\hat{y}_k = c_0 + c_1 x_1 + \ldots + c_k x_k. \tag{1}$$

For events like thunderstorms, heavy precipitation or strong wind speed, calibration of event occurrence or threshold exceedance probability is performed using multiple logistic regression. For this an estimate

$$\hat{y}_k = \frac{1}{1 + e^{-(c_0 + c_1 x_1 + \ldots + c_k x_k)}} \tag{2}$$

of the predictand $y$ is determined using a maximum likelihood approach. The predictand $y$ now is a binary variable that is 1 in case the event was observed and 0 if not, whereas the estimate $\hat{y}_k$ is considered a probability that takes values from 0 to 1. Logistic regression (Hosmer et al., 2013, e.g.) is a classical approach for probabilistic postprocessing.

Details of the implementation of linear and logistic regression are presented in Secs. 3.1 and 3.2, respectively. Especially for probabilities of strong and extreme wind gusts a global regression is applied that is presented in Sec. 3.3. For an introduction to MOS in general we refer to Glahn and Lowry (1972); Wilks (2011); Vannitsem et al. (2018).

## 3.1 Optimisation and interpretation by linear regression

Ensemble-MOS derives altogether some 150 predictands from synoptic observations (for precipitation also gauge adjusted radar products can be used) for statistical modelling. For the speeds of wind gusts and for precipitation amounts individual predictands for various reference periods (e.g. 1-hourly, 3-hourly, 6-hourly and longer) are defined. As usual, these predictands are modelled by individual linear regressions. The resulting statistical estimates are added to the list of available predictors for subsequent regressions during postprocessing. They are selected as predictors especially for probabilities that the speeds of wind gusts or precipitation amounts exceed predefined thresholds within the corresponding time frames (i.e. 1 h, 3 h, etc.).

In order to estimate the error of the current forecast, an error predictand

$$y^e = |\hat{y}_k - y| \tag{3}$$

is defined as the absolute value of the residuum. The corresponding estimate $\hat{y}_k^e$ is defined according to Eq. (1). This error predictand can be evaluated as soon as the estimate $\hat{y}_k$ is available. The absolute value is preferred over the root mean square (RMS) of the residuum, since it shows higher correlations to many predictors and a better linear fitting. For Gaussian distributions with density $\varphi_{\mu,\sigma^2}$ the absolute error $e$ (or mean absolute deviation) of the distribution can be estimated from the standard deviation $\sigma$ as

$$e = \int_{-\infty}^{\infty} |x - \mu|\, \varphi_{\mu,\sigma^2}(x)\, dx = 2 \int_{0}^{\infty} x\, \frac{1}{\sqrt{2\pi}\sigma} e^{\frac{-x^2}{2\sigma^2}}\, dx = \sqrt{\frac{2}{\pi}}\, \sigma \approx 0.8\, \sigma \,. \tag{4}$$

For each predictand the most relevant predictors are selected from a predefined set of independent variables by stepwise regression. Statistical modelling is performed for each predictand, station, season, forecast run and forecast time individually in general. For rare events, however, nine zones with similar climatology are defined (e.g. coastal strip, north German plain, various height zones in southern Germany, high mountain areas, etc.) and the stations are clustered together in order to increase the number of observed events and the statistical significance of the training data. All stations of a cluster are modelled together for those events.

Most potential predictors are based on forecasts of the ensemble system, which are interpolated to the locations of the observation sites. Additional to the model values at the nearest grid point, also mean and standard deviation of the $6 \times 6$ and $11 \times 11$ surrounding grid points are evaluated and provided as medium and large-scale predictors, respectively. Moreover, also extra variables are derived from the NWP-model fields to be used as predictors, as e.g. potential temperature, various atmospheric layer thicknesses, rotation and divergence of wind velocity, dew-point spread and even special parameters, such as convective available potential energy (CAPE) and severe weather threat index (SWEAT). These variables are computed from the ensemble means of the required fields.

Statistical forecasts of the same variable of the last forecast step and also of other variables of the current forecast step can be used as well. For example, forecasts of 2 m temperature may use statistical forecasts of precipitation amounts of the same time step as predictors. The order of the statistical modelling and of the forecasting is relevant in such cases to make sure the required data is available.

Further predictors are derived from the latest observations that are available at the time when the statistical forecast is computed. Generally, the latest observation is an excellent projection for short-term forecasts up to about four to six hours, which is therefore added to the set of available predictors. Special care has to be taken to process these predictors for training, however. Only those observations can be used that are available at run time of the forecast. In case forecasts are computed for arbitrary locations apart from observation and training sites, these observations or persistency predictors have to be processed

in exactly the same way in training and forecasting. At locations other than observation sites, the required values need to be interpolated from the surrounding stations. As interpolation generally is a weighted average based on horizontal and vertical distance, it introduces smoothing and, with it, a systematic statistical change in the use of the observations. If the training is performed using observations at the stations and the forecasting is using interpolation, the statistical forecasts can be affected. As a remedy, Ensemble-MOS uses observations as persistence predictors for training that are interpolated from up to five

surrounding stations in exactly the same way as when computing the forecast at arbitrary locations, even if an observation at the correct location was available.

    Also special orographic predictors exist, like height of station or height difference between station and model at a specific location. In order to address model changes also indicators or binary variables are provided (see Sec. 3.4 for details). Altogether more than 300 independent variables are defined, from which up to ten predictors are selected for each predictand during

multiple regression.

    During stepwise regression, the predictor with the highest correlation with the predictand is selected first from the set of available independent variables. Next, the linear regression with the previously chosen set of predictors is computed and the next predictor with the highest correlation with the residuum is selected, and so on. Selection stops, if no further predictor exists with a statistically significant correlation according to a Student's t-test. The level of significance of the test is 0.18 divided by

the number of available independent variables. This division is used because of the high number of potential predictors. With a type I error of e.g. 0.05 and a number of 300 available predictors, 15 predictors on average would be selected randomly without providing significant information. The value 0.18 is found to be a good compromise in order to select a meaningful number of predictors and to prevent overfitting in this scenario.

    Table 1 lists the most important predictors for statistical forecasts of the maximal speed of wind gusts within $1\,\mathrm{h}$. The

relative weights are aggregated over 5472 equations, one for each cluster, season, forecast run, and forecast lead time. Note, that predictors that are highly correlated usually exclude each other to appear within one equation. Only the predictor with the highest correlation with the predictand is selected and supplants other correlated predictors that do not provide enough additional information according to the t-test.

    The MOS-equations are determined by stepwise regression for individual locations and, in case of rare events, for clusters.

In order to compute statistical forecasts on a regular grid, these equations need to be evaluated for locations apart from the training and observation sites. In case of rare events and cluster equations, the appropriate cluster is determined for each grid points and the equation of that cluster is used. The equations for individual locations are interpolated to the required grid point by linear interpolation of their coefficients. In all cases, the required values of the numerical model for these equations

**Table 1.** Predictors for statistical forecasts of the maximal speed of wind gusts within 1 h that have relative weights higher than 1%. The relative weights are aggregated for all stations, seasons, forecast runs, and forecast lead times. The ensemble mean of COSMO-D2-EPS is denoted by DMO (direct model output). Parentheses within the predictor names denote time shifts. For a time shift of -30 min, denoted as (-0:30), the predictors are interpolated based on values for the previous and the current forecast hour. The required statistical forecasts have to be evaluated in advance.

| predictor name | rel. weight [%] | description |
| --- | --- | --- |
| FF(-0:30)StF | 35.7 | statistical forecast of mean wind speed in 10 m height |
| FX1(-1)StF | 12.1 | statistical forecast of speed of wind gusts of the previous hour |
| VMAX_10M_LS | 8.9 | DMO of speed of wind gusts for a large surrounding area (mean of $11 \times 11$ grid points) |
| VMAX_10M | 8.6 | DMO of speed of wind gusts for next model grid point |
| FF_850(-0:30) | 4.8 | DMO of wind speed in 800 h Pa height |
| VMAX_10M_MS | 4.1 | DMO of speed of wind gusts for a medium surrounding area (mean of $6 \times 6$ grid points) |
| Oa_D_0.5 | 3.2 | latest observation of the speed of wind gusts |
| FF_10m(-0:30) | 2.8 | DMO of mean wind speed in 10 m height |
| StFT2m_T950 | 1.9 | statistical forecast of temperature difference between 2 m and 950 h Pa height (stability index) |
| Location-height | 1.6 | height of station |
| FF_1000(-0:30) | 1.1 | DMO of wind speed in 1000 h Pa height |

are evaluated for the exact location. Observations that are used as persistency predictors are interpolated from surrounding sites. In this way, gridded forecast maps can be obtained as displayed in Fig. 2 for wind gust probabilities (see Sec. 3.2 for probabilistic forecasts). For computational efficiency, the forecasts are initially computed on a regular grid of 20 km resolution and are downscaled thereafter to 1 km while taking into account the various height zones in southern Germany. The details of the downscaling are beyond the scope of the paper.

### 3.2 Calibration of probabilistic forecasts by logistic regression

Event probabilities are calibrated using logistic regression. Equation (2) is solved using a maximum likelihood approach. The likelihood function

$$P(y, c_0, \ldots c_k) = \prod_{i=1}^{n} (\hat{y}_k^i)^{y^i} (1 - \hat{y}_k^i)^{1-y^i} \tag{5}$$

expresses the probability that the predictand $y$ is realised given the estimate $\hat{y}_k$ via the coefficients $c_0, \ldots, c_k$ (and by now with fixed predictors $x_1, \ldots, x_k$), with $n$ being the time dimension (sample size) and $i$ the time index. The predictand $y$ of an event probability is binomially distributed, its time-series values are defined as 1 in case the event was observed and 0 if not, whereby conditional independence is assumed in Eq. (5).

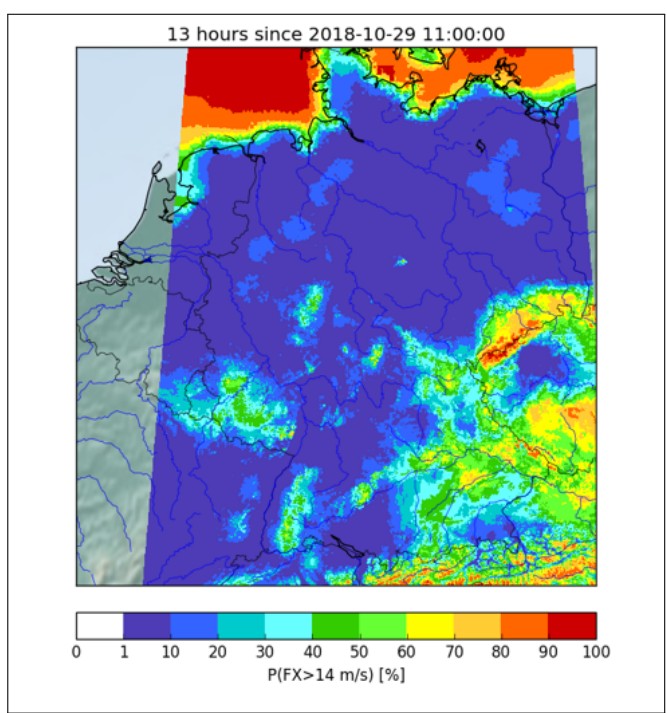

**Figure 2.** Probabilities for wind gusts higher than $14\,\mathrm{m\,s}^{-1}$ on a regular $1\,\mathrm{km}$ grid over Germany, $13\,\mathrm{h}$ forecast lead time from Ensemble-MOS for COSMO-DE-EPS from 29 October 2018

It is mathematically equivalent and computationally more efficient to maximise the logarithm of the likelihood function

$$\ln(P(y, c_0, \ldots c_k)) = \sum_{i=1}^{n} \left( y^i \ln(\hat{y}_k^i) + (1 - y^i) \ln(1 - \hat{y}_k^i) \right) . \tag{6}$$

This maximisation is implemented by calling the Routine `G02GBF` of the NAG-Library in FORTRAN 90, see Numerical Al-
gorithms Group (1990). The resulting fit of the estimate $\hat{y}_k$ can be evaluated by the deviance

$$D_k = -2\ln\left(P(y, c_0, \ldots, c_k)\right), \tag{7}$$

which is a measure analogous to the squared sum of residua in linear regression.

The selection of predictors is again performed stepwise. Initially, the coefficient $c_0$ of the null model $\bar{y} = \frac{1}{1 + e^{-c_0}}$ that fits
the mean of the predictand is determined and the null deviance $D_0 = -2\ln(P(y, c_0))$ is computed. The coefficient $c_0$ is
often called the intercept. Starting from the null model the predictor that is selected first is the one that shows the smallest
deviance $D_1 = -2\ln\left(P(y, c_0, c_1)\right)$. The difference $D_1 - D_0$ is $\chi_1^2$-distributed with 1 degree of freedom and is used to check
the statistical significance of the predictor. This check replaces the t-test in linear regression and uses the same statistical level.
If the predictor shows a significant contribution, it is accepted and further predictors are tested based on the new model in the
same way. Otherwise, the predictor is rejected and the previous fitting $\hat{y}_{k-1}$ is accepted as the final statistical model.

As a rule of thumb, for each selected predictor in the statistical model at least ten events need to be captured within the observation data (*one in ten rule*) to find stable coefficients. For example, with only 30 events in the training set, the number of predictors should be restricted to three. This rule is critical especially for rare events such as extreme wind gusts or heavy precipitation.

Since testing all candidate predictors from the set of about 300 variables by computing their deviances is very costly, the score test (Lagrange multiplier test) is actually applied to Ensemble-MOS. Given a fitted logistic regression with $k-1$ selected predictors, the predictor is chosen next as $x_k$, that shows the steepest gradient of the log-likelihood function Eq. (6) in an absolute sense when introduced, normalised by its standard deviation $\sigma_{x_k}$, i.e.

$$\frac{1}{\sigma_{x_k}}\left|\frac{\partial \ln\big(P(y,c_0,\ldots,c_k)\big)}{\partial c_k}\bigg|_{c_k=0}\right| = \left|\sum_{i=1}^{n}\big(y^i - \hat{y}^i_{k-1}\big)\frac{x^i_k}{\sigma_{x_k}}\right|. \tag{8}$$

This equation results from basic calculus including the identity $\frac{\partial \hat{y}_k}{\partial c_k} = \hat{y}_k(1-\hat{y}_k)\,x_k$. The right hand side of Eq. (8) is basically the correlation of the current residuum to the new predictor. The score test thus results in the same selection criterion as applied to stepwise linear regression. Once the predictor $x_k$ is selected, the coefficients $c_0,\ldots,c_k$ are updated to maximise Eq. (6).

### 3.3  Global logistic regression of wind gust probabilities

For extreme events, the number of observed occurrences can still be too small to derive stable MOS-equations, although time series of several years have been gathered and the stations are clustered within climatologic zones in Germany. The eight warning thresholds of DWD for wind gusts range from $12.9\,\mathrm{m\,s^{-1}}$ ($25.0\,\mathrm{kn}$, proper wind gusts) up to $38.6\,\mathrm{m\,s^{-1}}$ ($75.0\,\mathrm{kn}$, extreme gales), whereas the maximal observed speed of wind gusts in the training data for a cluster in the northern German plains is only $25.4\,\mathrm{m\,s^{-1}}$. Especially for probabilities of extreme wind gusts global logistic regressions are developed, that use events at the coastal strip or at mountains in southern Germany and allow for meaningful statistical forecasts of extreme events also in climatologically calm areas. The statistical forecasts of the continuous speed of wind gusts are used for these logistic regressions as the only predictors. They are modelled by stepwise linear regression for each station individually, as described in Sec. 3.1. In this way, rare occurrences of extreme events are gathered globally, while concurrently a certain degree of locality is maintained.

The locally optimised and unbiased forecasts of wind gust speeds are excellent predictors for wind gust probabilities. The logistic regressions according to Eq. (2) with $k=1$ fit the distributions of observed wind gusts quite well as shown in Fig. 3 for threshold $t = 13.9\,\mathrm{m\,s^{-1}}$ ($27.0\,\mathrm{kn}$) and forecast lead times of $1\,\mathrm{h}$ and $7\,\mathrm{h}$, respectively.

The statistical modelling of wind gust probabilities is performed for each threshold $t$ individually and is described in the following. The logistic regressions represent logistic distributions with mean $\mu_t = -\frac{c_0}{c_1}$ and variance $\sigma_t^2 = \frac{\pi^2}{3c_1^2}$, which are computed for various lead times $h$ and are listed in Tab. 2. The expectations $\mu_t$ are slightly smaller than the threshold $t = 13.9\,\mathrm{m\,s^{-1}}$, almost independently of lead time. The reason is that for given statistical forecasts of wind gusts the distribution of observations is almost Gaussian, see Fig. 4, albeit a little left skewed with a small number of very weak wind observations. The

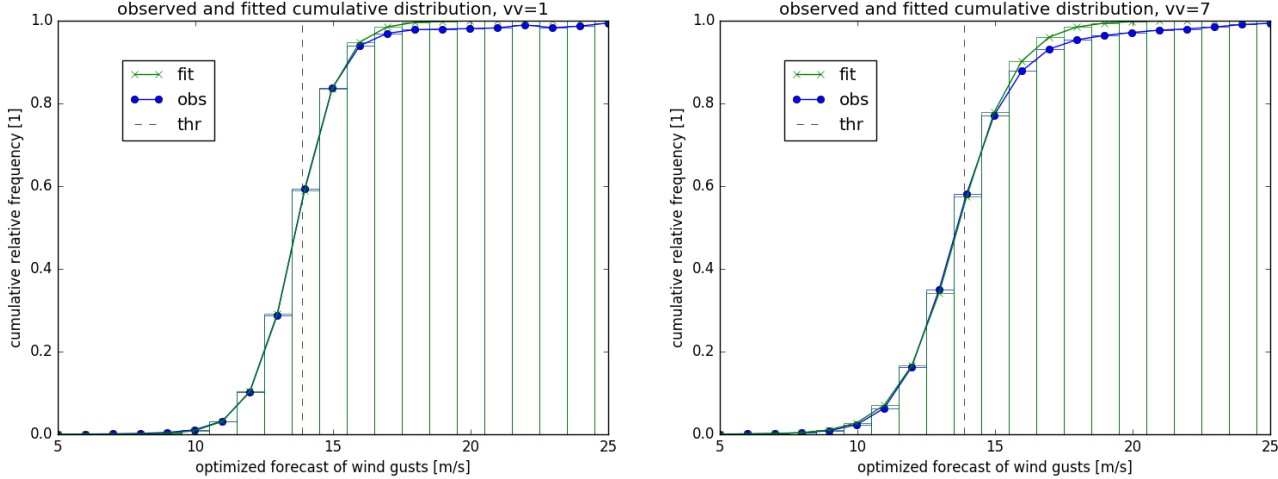

**Figure 3.** Observed cumulative distributions of wind gusts exceeding threshold $13.9\,\mathrm{m\,s^{-1}}$ (blue) and fit of logistic distribution (green) depending on statistically optimised forecasts of wind gusts for forecast lead times 1 h (left) and 7 h (right). The threshold is dashed.

**Table 2.** Parameters of fitted logistic distributions as shown in Fig. 3 for various forecast lead times $h$, with coefficients of logistic regressions $c_0$ and $c_1$ and resulting means $\mu_t$ and standard deviations $\sigma_t$ for threshold $t = 13.9\,\mathrm{m\,s^{-1}}$. Estimated uncertainties are given in brackets.

| $h$ | $c_0$ | $c_1$ | $\mu_t$ | $\sigma_t$ |
|---|---|---|---|---|
| 1 | -17.74 ($\pm$0.17) | 1.30 ($\pm$0.01) | 13.68 ($\pm$0.01) | 1.40 ($\pm$0.01) |
| 4 | -13.76 ($\pm$0.11) | 1.01 ($\pm$0.01) | 13.66 ($\pm$0.01) | 1.80 ($\pm$0.02) |
| 7 | -13.32 ($\pm$0.11) | 0.98 ($\pm$0.01) | 13.66 ($\pm$0.01) | 1.86 ($\pm$0.02) |
| 10 | -13.00 ($\pm$0.10) | 0.95 ($\pm$0.01) | 13.66 ($\pm$0.01) | 1.91 ($\pm$0.02) |
| 16 | -12.51 ($\pm$0.10) | 0.92 ($\pm$0.01) | 13.66 ($\pm$0.01) | 1.98 ($\pm$0.02) |

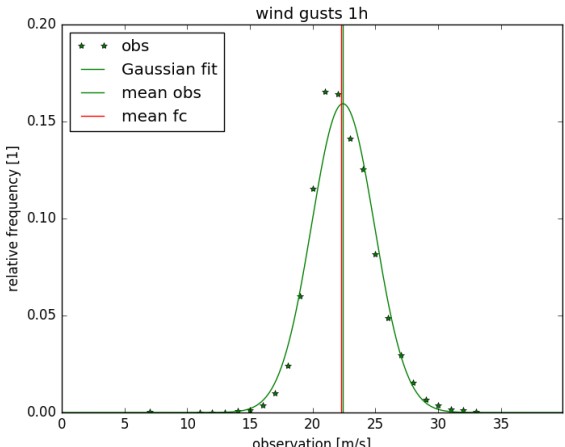

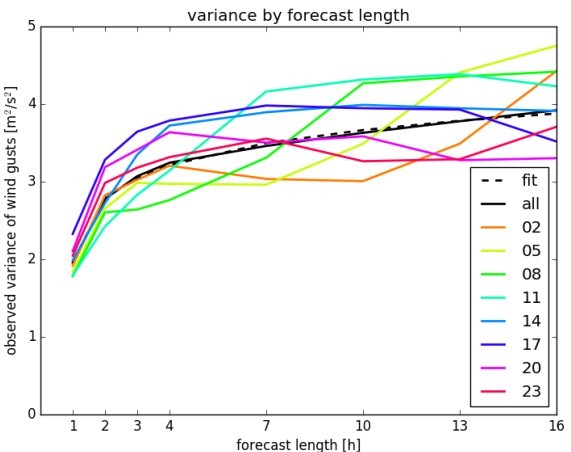

**Figure 4.** Distribution of wind gust observations from 2011 to 2016 for 178 synoptic stations for cases where statistical forecast (fitted for that period) is between $21\,\mathrm{m\,s^{-1}}$ and $25\,\mathrm{m\,s^{-1}}$. Lead time is $1\,\mathrm{h}$. Gaussian fit and mean of observations (green), mean of forecasts (red)

**Figure 5.** Variances $\sigma_t^2$ of logistic distributions fitted to cumulative distributions of observed wind gusts for threshold $t = 13.9\,\mathrm{m\,s^{-1}}$ depending on forecast lead time. Individual runs of Ensemble-MOS for COSMO-DE-EPS-MOS starting at $02\,\mathrm{UTC}$, $05\,\mathrm{UTC}, \ldots, 23\,\mathrm{UTC}$ in colours, mean of all runs in black, fitted parameterisation of variances dashed in black

standard deviations $\sigma_t$ increases with forecast lead time reflecting the loss of accuracy of the statistical forecasts. Consequently, the graph of the cumulative distribution function in Fig. 3 is more tilted for a forecast lead time of $7\,\mathrm{h}$ than for $1\,\mathrm{h}$.

Figure 5 shows fitted variances $\sigma_t^2$ of the eight individual forecast runs of Ensemble-MOS for COSMO-DE-EPS and their mean depending on lead time. In order to reduce the number of coefficients and to increase consistency and robustness of the forecasts, the variance $\sigma_t^2$ is parameterised depending on forecast lead time $h$ by fitting the function

$$\sigma_t^2(h) = c_t \log(a_t h + b_t) \tag{9}$$

with its parameters $a_t$, $b_t$, and $c_t$ for threshold $t$.

The fitted expectations and variances show weak dependencies on the time of the day and are neglected. The logistic regressions of wind gust probabilities thus can be expressed for each threshold $t$ by the mean $\mu_t$ and $\sigma_t$ for all start times of the forecasts in the same way.

Even for very rare gales of $38.6\,\mathrm{m\,s^{-1}}$ more than 130 events are captured using six years of training data when modelling all stations and forecast runs together, which is sufficient for logistic regression. Training for these extreme events is based mainly on coastal and mountain stations, but the statistical regressions are applied to less exposed locations in calmer regions as well. Small threshold probabilities will be predicted for those locations in general. However, meaningful estimations will be generated once the statistical forecasts of local wind speed rises induced by the numerical model.

### 3.4 Specific issues and caveats of MOS

Ensemble-MOS optimises and calibrates ensemble forecasts using synoptic observations. Being a statistical method, it is vulnerable to systematic changes in input data, since it assumes that errors and characteristics of the past persist in future. An important part of the input are observations, whose measurement instruments sometimes change. It is recommended to use quality-checked observations in order to avoid the use of defective values for training. Especially observation sites that are automatised need to be screened. Furthermore, numerical models change with new versions and updates that can affect statistical postprocessing, as further discussed in Sec. 3.4.1.

Although statistical forecasts generally improve the model output when verified against observations, the results are not always consistent in time, space and between the forecast variables (e.g. between temperature and dew point), if they are optimised individually. This issue is addressed in Sec. 3.4.2.

### 3.4.1 Model changes

Statistical methods like Ensemble-MOS detect systematic errors and deficiencies of NWP-models during a past training period in order to improve topical operational forecasts. Implicitly it is assumed, that the systematic characteristics of the NWP-models persist. Note that multiple regressions not only correct for model bias, but also for conditional biases that depend on other meteorological variables. Multiple regressions are more vulnerable to model changes than simple regressions, therefore. Systematic changes in NWP-models can affect statistical forecasts, even if the NWP-forecasts are objectively improved as confirmed by verification. Given that the statistical modelling provides unbiased estimations, any systematic change in NWP-model predictors will reflect in biases in the statistical forecasts. The resulting biases depend on the magnitudes of the changes of the predictors and on their weights in the MOS-equations.

One remedy for jumps in input data is the use of indicator (binary) predictors. These predictors are related to the date of the change of the NWP-model and are defined as 1 before and 0 after. When they are selected during stepwise regression, they account for sudden jumps in the training data and can prevent the introduction of unconditional biases in the statistical forecasts. Conditional biases depending on other forecast variables, however, are not corrected.

In order to process extreme and very rare events for weather warnings, long time series of seven years of data for COSMO-D2-EPS have been gathered at the time of writing. Hence, the time series are subject to a number of model changes. A significant model upgrade from COSMO-DE to COSMO-D2, including an increase of horizontal resolution from $2.8\,\mathrm{km}$ to $2.2\,\mathrm{km}$ and an update of orography, took place in May 2018. Since reforecasting of COSMO-D2-EPS for more than one year was technically not possible, the existing COSMO-DE-EPS database was used further and extended with reforecasts of COSMO-D2-EPS of the year before operational introduction. However, statistical experiments using these reforecasts of COSMO-D2-EPS (and the use of binary predictors, see above) revealed only insignificant improvements compared to training with data of COSMO-DE-EPS only. For rare events, longer time series are considered more important than the use of unaltered model versions.

### 3.4.2 Forecast consistency

As weather warnings are issued for a certain period of time and a specified region, continuity of probabilistic forecasts in time
and space is important. It should be accepted, however, that maps of probabilistic forecasts do not comply with deterministic
runs of numerical models, as probabilistic forecasts are smoothed according to forecast uncertainty. For example, there are
hardly convective cells in probabilistic forecasts, but rather areas exist where convection might occur with a certain probability
within a given time period.

The statistical modelling of Ensemble-MOS is carried out for each forecast variable, forecast lead time and location inde-
pendently and individual MOS-equations are derived. For rare meteorological events clusters of stations are grouped together
that are similar in climatology in order to derive individual cluster equations. This local and individual fitting results in optimal
statistical forecasts for the specific time, location and variable as measured with the RMSE compared to observations. However,
it does not guarantee that obtained forecast fields are consistent in space, time or between variables.

In forecast time, spurious jumps of statistical forecasts can appear and variables with different reference periods usually do
not match. For example, the sum of twelve successive one-hourly precipitation amounts would not equal the corresponding 12-
hourly amount, if the latter is modelled as an individual predictand. Statistical forecasts of temperature cannot be guaranteed to
exceed those of dew point. Maps of statistical forecasts show high variability from station to station and unwanted anomalies
in case of cluster equations. Cluster edges turn up and it may appear that there are higher wind gusts in a valley than on a
mountain nearby, in cases where the locations are arranged in different clusters, for example. For consistency in time and
space the situation can be improved by using the same equations for several lead times and for larger clusters or by elaborate
subsequent smoothing. However, forecast quality for a given space and time will be degraded consequently. For consistency
between all forecast variables multivariate regressions are required that model the relevant predictands simultaneously.

From the point of view of probabilistic forecasting, however, statistical forecasts are random variables with statistical dis-
tributions, although commonly only their expectations are considered as *the* statistical forecast. In case forecast consistency
is violated from a deterministic point of view, this is not the case if statistical errors are taken into account. The statistical
forecasts remain valid as long as the probability distributions of the variables overlap. As this is a mathematical point of view,
the question remains, how to communicate this nature of probabilistic forecasts to the public or traditional meteorologists in
terms of useful and accepted products.

## 4 Results

Evaluation of Ensemble-MOS for COSMO-DE-EPS and ECMWF-ENS is provided in the following. Although Ensemble-
MOS of DWD provides statistical forecasts of many forecast variables that are relevant for warnings, evaluation is focused on
wind gusts for COSMO-DE-EPS and on temperature for ECMWF-ENS in order to limit the scope of the paper.

## 4.1 Evaluation of Ensemble-MOS for COSMO-DE-EPS

Verifications of the continuous speed of wind gusts are presented in Figs. 6 - 8 by various scatter diagrams including forecast means (solid line) and their standard errors (dashed lines). Figures 6 (right) and 7 (right) show the statistical fit of the speed of wind gusts against synoptic observations during a training period of six years of Ensemble-MOS for COSMO-DE-EPS for lead times of 1 h and 6 h, respectively. The fit is almost unbiased for all forecast speed levels. The raw ensemble means show overforecasting for high wind gusts (same Figs., left) and the standard errors are considerably larger. If no overfitting occurs, out-of-sample forecasts are expected to behave accordingly, which is verified in Fig. 8 (right) for a test period of three months (at least for wind gusts up to about $20\,\mathrm{m\,s^{-1}}$).

Ensemble-MOS can predict its own current forecast errors by using error predictands according to Eq. (3). Forecasts of the absolute errors of the speeds of wind gusts are related to observed errors in Fig. 9 (right). The biases are small, although individual observed errors are much larger than their predictions. The absolute errors of the ensemble mean versus ensemble spread (normalised to absolute error) strongly underestimate the observed errors of the ensemble mean, see Fig. 9 (left). This is another example for underestimated dispersion of COSMO-DE-EPS as shown in Fig. 1 for precipitation.

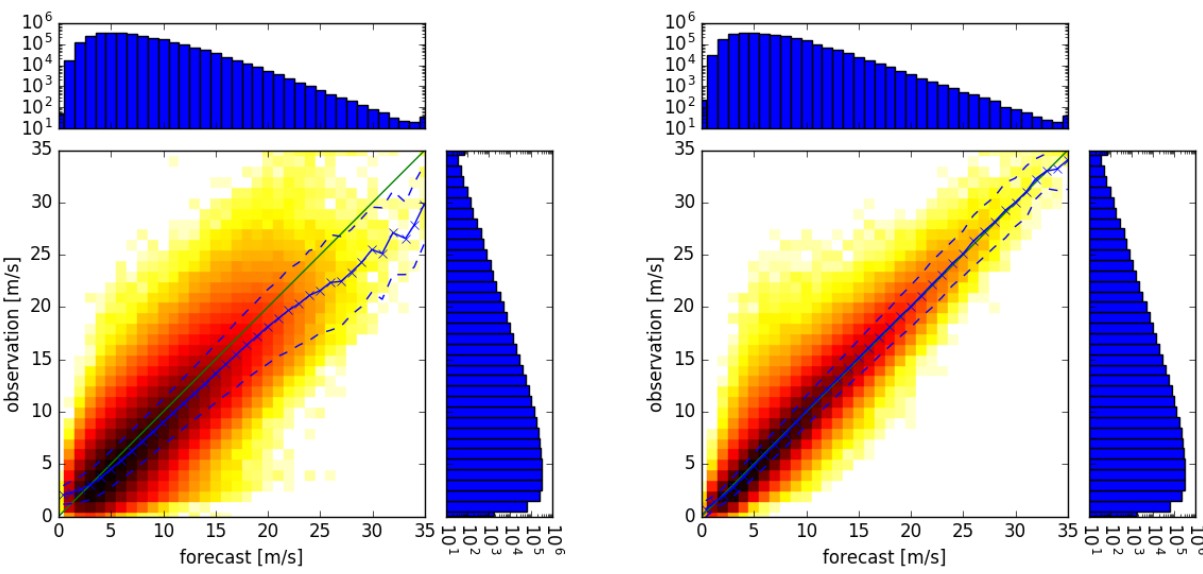

**Figure 6.** Scatter plots of ensemble means of 3 h forecasts of the speeds of wind gusts of COSMO-DE-EPS versus observations (left) and corresponding statistical fits of 1 h forecasts of Ensemble-MOS versus same observations (right). Means of observations (solid) and confidence intervals (means +/- standard deviations, dashed) are shown. 6 years of data (2011-2016) are used, number of cases are given by histograms.

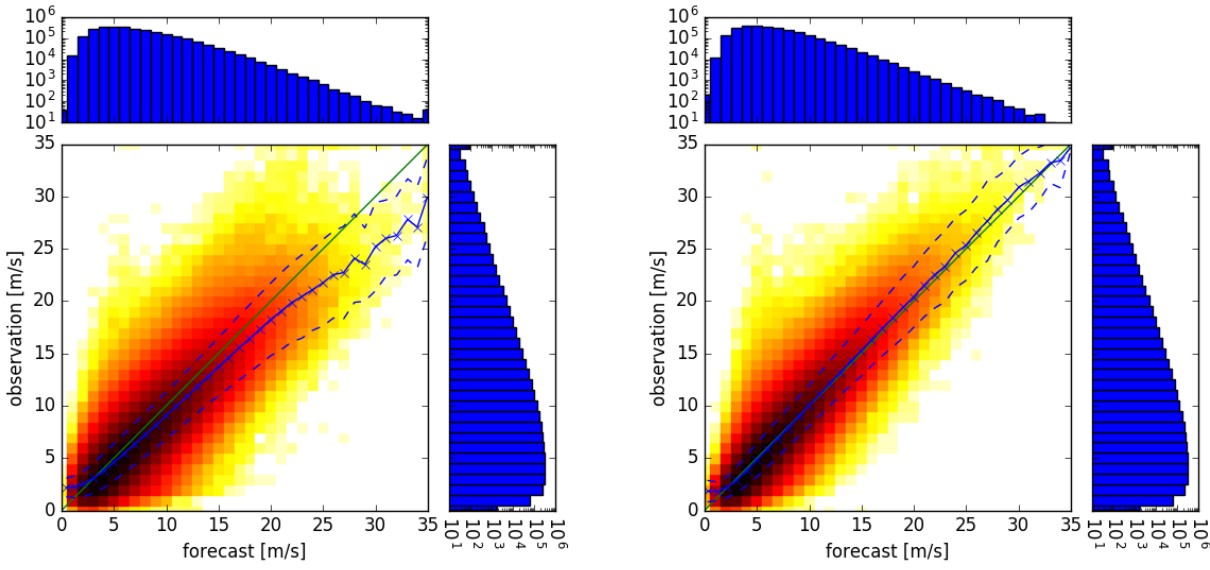

**Figure 7.** As Fig. 6, but for 8 h forecasts of COSMO-DE-EPS (left) and 6 h forecasts of Ensemble-MOS (right)

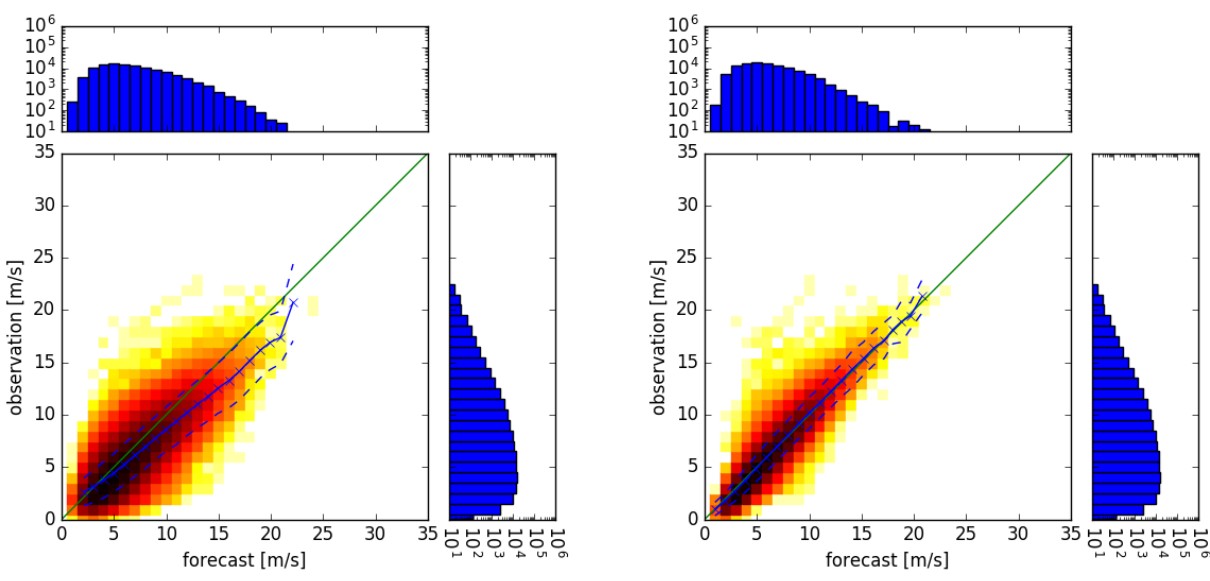

**Figure 8.** As Fig. 6, but for three months of data (May-July 2016) and forecasts of COSMO-DE-EPS (left) and Ensemble-MOS (right). Data of this period were not used for training.

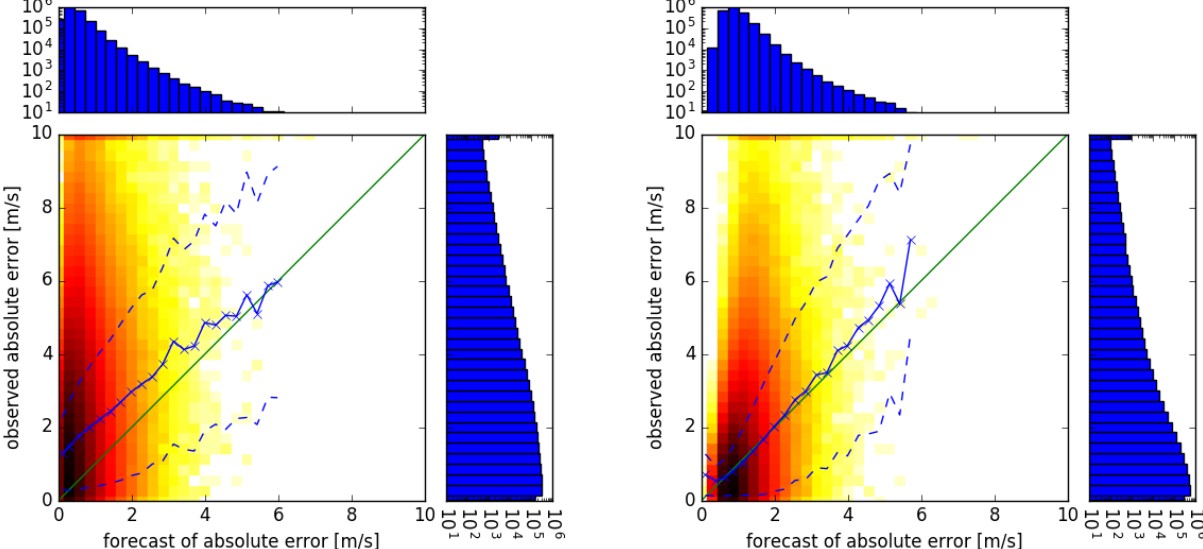

**Figure 9.** Scatter plots of 3 h forecasts of the absolute errors of COSMO-DE-EPS forecasts of wind gusts speeds (estimated as ensemble standard deviations*0.8) versus observed absolute errors of the ensemble means (left) and corresponding 1 h error forecasts of Ensemble-MOS versus observed absolute errors of Ensemble-MOS (statistical fit of training period, right). Means of observed absolute errors (solid) and confidence intervals (means +/- standard deviations, dashed) are shown. 6 years of data (2011-2016) are used.

The statistical forecasts of the speeds of the wind gusts are excellent predictors for the probabilities that certain warning thresholds are exceeded. This is demonstrated by the fits of the observed distributions by logistic regression as shown in Fig. 3. The global logistic regression presented in Sec. 3.3 is prepared for extreme and rare events, nevertheless it is applicable to lower thresholds as well. The reliability diagram Fig. 10 shows well-calibrated probabilities for wind gusts exceeding $7.7\,\mathrm{m\,s^{-1}}$ for a zone in the northern German plains with calmer winds in climatology. The COSMO-DE-EPS shows strong overforecasting in these situations.

## 4.2 Evaluation of Ensemble-MOS for ECMWF-ENS

In order to motivate the use of Ensemble-MOS for ECMWF-ENS a study has been carried out with a restricted set of model variables of TIGGE, see Sec. 2.3. Training is based on ensemble data and corresponding observations from 2002 - 2012, whereas statistical forecasting and verification is performed for 2013, see Hess et al. (2015) for details.

Results for 2 m temperature forecasts are shown in Fig. 11, which illustrates essential improvements of postprocessed forecasts of Ensemble-MOS compared to raw ensemble output. The statistical forecast (blue) not only improves the raw ensemble mean (red), it also outperforms the high resolution ECMWF-IFS (this data has not been used for training). Also the statistical estimation of Ensemble-MOS of its own errors (pink), see Sec. 3.1, is more realistic over the first few days than the estimate of

the ensemble mean errors by the ensemble spread (yellow). Improvements of ECMWF-ENS with Ensemble-MOS were also obtained for 24 h precipitation and cloud coverage.

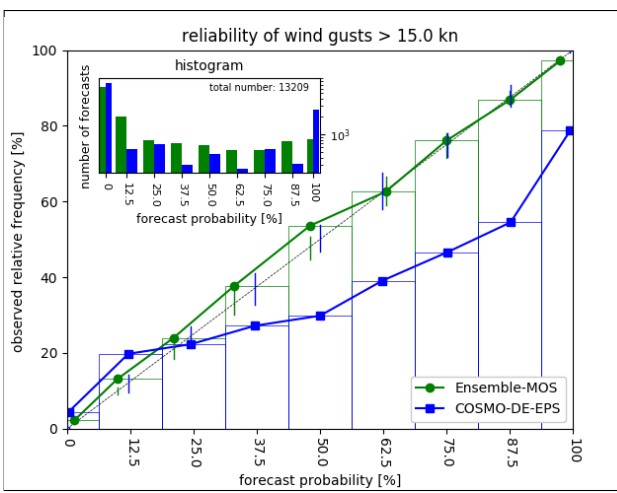

**Figure 10.** Reliability diagram for probabilities of wind gusts exceeding $7.7\,\mathrm{m\,s^{-1}}$ (15.0 kn). Probabilistic forecasts of Ensemble-MOS with lead time of 6 h (green) and corresponding relative frequencies of COSMO-DE-EPS with lead time of 8 h (blue). Verification is done for three months of data (May-July 2016) and 18 stations in Germany at about N52° latitude including Berlin for example. Vertical lines are 5%-95% consistency bars according to Bröcker and Smith (2006).

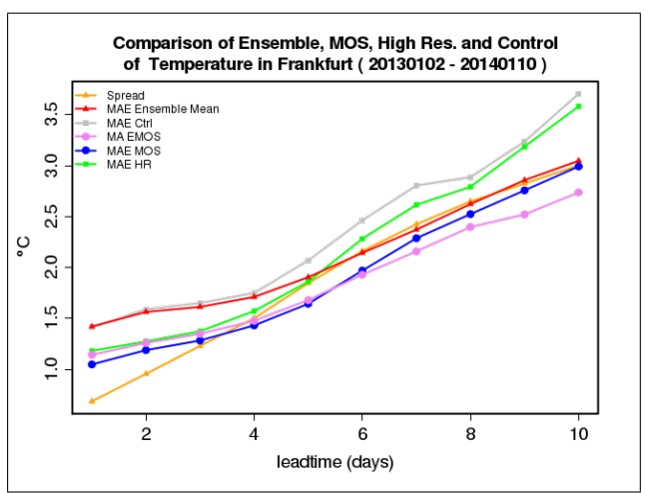

**Figure 11.** Mean absolute error (MAE) of 2 m temperature forecast and error estimations depending on forecast lead time. Spread (yellow): spread of ECMWF-ENS (normalised to MAE); MAE Ensemble Mean (red): MAE of mean of ECMWF-ENS; MAE Ctrl (grey): MAE of ECMWF-ENS control run; MA EMOS (pink): Ensemble-MOS forecast of its own absolute errors (see Eq. (3): estimations of MAE EMOS, blue); MAE MOS (blue): MAE of Ensemble-MOS for ECMWF-ENS; MAE HR (green): MAE of high resolution ECMWF-IFS

## 5  Conclusions

This paper describes the Ensemble-MOS system of DWD, which is set up to postprocess the ensemble systems COSMO-D2-EPS and ECMWF-ENS with respect to severe weather to support warning management. MOS in general is a mature and sound method and in combination with logistic regression, it can provide optimised and calibrated statistical forecasts. Stepwise multiple regression allows reducing conditional biases that depend on the meteorological situation, which is defined by the selected predictors. The setup of Ensemble-MOS to use ensemble mean and spread as predictors is computationally efficient and simplifies forecasting of calibrated event probabilities and error estimates on longer forecast lead times. Ensemble-MOS is operationally applicable with regard to its robustness and computational costs and runs in trial mode in order to support warning management at DWD.

The ensemble spread is less often detected as an important predictor as might be expected, however. One reason is that the spread actually carries less information about forecast accuracy as originally intended. It is often too small and too steady to account for current forecast errors. Another reason is that some forecast variables correlate with their own forecast errors (e.g. precipitation and wind gusts). If the ensemble spread does not provide enough independent information, it is not selected additionally to the ensemble mean during stepwise regression. Currently, only ensemble mean and spread are provided as predictors for Ensemble-MOS. The implementation of various ensemble quantiles as additional predictors is technically straightforward and could improve the exploitation of the probabilistic information of the ensemble.

Statistical forecasts of the speed of the wind gusts are excellent predictors for probabilities that given thresholds are exceeded and are used as predictors within logistic regressions. The same approach could be advantageous for probabilities of heavy precipitation as well, where estimated precipitation amounts would be used as predictors.

An important further step in probabilistic forecasting is the estimation of complete (calibrated) distributions of forecast variables rather than forecasting only discrete threshold probabilities. For wind gusts with Gaussian conditional errors as shown in Fig. 4 this seems possible but certainly requires additional research.

With its inherent linearity (also in the case of logistic regressions there are linear combinations of predictors only) MOS has its restrictions in modelling, but supports traceability and robustness, which are important features in operational weather forecasting. Therefore, MOS is considered a possible baseline for future statistical approaches based on neural networks and machine learning that allow for more general statistical modelling. Many of the statistical problems will remain however, such as finding suitable reactions to changes in the NWP-models, (deterministic) consistency and the definition of useful probabilistic products (see Sec. 3.4.2) and the verification of rare events. In all cases, training data is considered of utmost importance, including the NWP-model output, as well as quality-checked historic observations.

*Author contributions.* Conceptual design of Ensemble-MOS, enhancement of software for probabilistic forecasting of ensembles (including logistic regression), processing of forecasts, verification and writing was done by the author.

*Competing interests.* The author declares that he has no conflicts of interest.

*Acknowledgements.* The author thanks two anonymous reviewers and the editor for their constructive comments, which helped to improve the structure and clarity of the manuscript. Thanks to James Paul for improving the use of English.

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
