# Peer review of "Statistical Postprocessing of Ensemble Forecasts for Severe Weather at Deutscher Wetterdienst"

_Nonlinear Processes in Geophysics, 2019_

## Referee Comment (RC1) · Anonymous Referee #1 · 12 Feb 2020

The article describes DWD's system to postprocess ensemble forecasts using the classical techniques linear and logistic regression with a focus on severe weather. While there is nothing new in the statistical methodology, the most interesting part lies in how the methods are applied to the data. The descriptions of the selection and design of predictors, how predictions are made at any location, how to deal with model changes, rare events etc. are all of interest to readers involved in operational forecasting. Possible weaknesses of the system could be discussed in more detail along with prospects of using more modern statistical methods. Please see below for more specific comments.

**Specific comments**

- line 63: Not sure I understand what the meaning of "... in order to avoid underestimation of forecast errors on longer time scales"
- line 106: As I understand, the combination is done after calibrating each model, but there seems to be no information about how this is done, apart from the citation of Reichert et al. (2015). Maybe a short paragraph on this interesting issue could be added to section 2.
- line 110. I find "Ensemble-MOS" a bit confusing here. Maybe it could be added that it is the name of DWD's system rather than referring to all "EMOS" methods.
- line 135, Fig. 4d: Why do you use absolute errors instead of squared errors which is both more precise and commonly used?
- Fig. 4d: What is meant by "forecast of absolute error (m/s)"? Do you make predictions of the absolute errors directly or do you compute the absolute error after the gust forecasts are made?
- line 140: Is the upscaled product an ensemble of 10x10x20 members or 20 members?
- line 145: What kind of bias metric is this, given that the forecasts are probabilities exceeding 0.1 mm/hour?
- lines 160-165: Here results using Ensemble-MOS are presented before Ensemble-MOS is described in section 3. This is unfortunate and can easily cause misunderstandings, e.g. it is not obvious that Ensemble-MOS produce deterministic forecasts. Please consider moving this paragraph.
- line 167: As I understand, it is also used to make deterministic forecasts for continuous variables. Since this is a bit unusual, it should be stated explicitly somewhere, the sooner the better.
- lines 172-174: An alternative explanation: linear regression applied individually to an ensemble results in an ensemble of conditional means which is not the same as the conditional distribution of the predictand/observation given the predictors. In practice, the more uncertainty there is, the more will this adjusted ensemble underestimate the uncertainty.
- line 193: Is it predictands or predictors? The sentence is unclear.
- line 195: Would "binary predictands" be more precise than "probability predictands"?
- lines 195-200: Are then predictions from the linear regression interpreted as probabilities of exceeding the given threshold? Is there any statistical justification of

the approach? Any references? How good is this approach compared to logistic regression? Maybe this paragraph should be expanded somewhat.

- line 202: replace predictor by predictand?
- line 237: I do not understand the sentence "Modeling this error ...". Do you mean that separate linear regression with the absolute error as predictand are made to predict the absolute error?
- line 240: Any reference or better?
- line 243: Do you mean the regression coefficients are interpolated separately?
- line 254: I would claim that logistic regression is no longer state of the art, at least in the research literature.
- Equation 3: Maybe it could be added that conditional independence is assumed
- lines 276-277: Not sure I understand this sentence. E.g. if there are 30 events in the training set, you would restrict the number of predictors to three?
- line 299: These are deterministic predictions of gusts, or?
- lines 300-301: Have you considered adding predictors describing the sites, e.g. elevation, exposure, land type etc.?
- lines 307: please consider using "deterministic wind gust forecasts"
- line 311: have you investigated whether the globally fitted logistic regression make well-calibrated probabilities for each station, say, for the lower thresholds to get sufficient data?
- lines 312-326: these paragraphs need more accurate explanations. What is the relation between logistic regression and the logistic distribution? What is the fitted logistic distribution used for? What is the purpose?
- line 400-402: what about raw ensemble probabilities for logistic regression?

**Technical corrections**

- line 40: Is "landmarks" appropriate?
- line 57: Should it be postprocessing?
- line 60: "coupula" should be "copula"
- line 62-63: maybe just "... tailored to postprocessing ensembles for extreme and rare events"?
- Figure 2: What is "MA EMOS"? Not easy to understand "Ensemble-MOS forecast of absolute error of Ensemble-MOS" either.
- line 202: each predictand(?), ...
- Equation 4: Parentheses are missing(?). Summation should be over both terms, I presume.
- line 299: stepwise linear regression(?)
- figure 4: text in the plots is too small. What are the dashed lines?
- figure 6, caption: "is between"

---

## Referee Comment (RC2) · Anonymous Referee #2 · 27 Feb 2020

General comments

The manuscript describes the so-called Ensemble-MOS as operated at Deutscher Wetterdienst. The statistical methods applied in Ensemble-MOS are quite standard. Ensemble-MOS is based on univariate multiple linear and logistic regression. The interesting aspects of this manuscript is the operational implementation, which uses a large amount of predictors and predictants and a stepwise variable selection. The manuscript contains interesting aspects of how to apply postprocessing in an operational setting. Nevertheless, an extensive revision of the manuscript is necessary, especially with regard to structure and language. Further, several aspects should be

examined more closely.

Specific comments

1. The general readability of the manuscript should be greatly improved. A clear separation in sections of (i) general aspects in postprocessing with literature, (ii) observations and forecasts, (iii) statistical methodology, (iv) DWD specific implementation and (v) results would largely increase the readability of the manuscript.

2. A thorough revision of the English language is necessary. Language editing should also pay attention to the clarity in terminology.

3. Since the manuscript employs standard statistical methods, the description of the statistics may be shortened. On the other hand, the description of the non-standard approach (why?) for wind gusts (section 3.3) is very confusing.

4. The results and particularly the figures should be discussed more closely. The order the figures are mentioned in the article must be consistent with the figure number.

5. The title emphasizes on severe weather. The manuscript describes both, and severe weather is restricted to wind gusts. So I suggest investigate postprocessing for severe weather more closely. No results are given with respect to the warning issue. No scores are presented, and no comparison is provided of the performance of the not-postprocessed ensemble and the benefit obtained with postprocessing.

6. No results a given with respect to the selected predictors, particularly for severe weather it might be interesting, which variables are most informative.
7. More details could be provide on the effect of model changes. In line 354ff you discuss the inclusion of an indicator. Do you detect any sudden changes? More details may also be provided, how the change from COSMO-DE-EPS to COSMO-D2-EPS affects the postprocessing (you only mention small improvements in COSMO-D2-EPS in lines 363-365).

Minor comments and typos

L. 2 "support warning decision management" $\rightarrow$ "support the warning decision management"

L. 8 "capture sufficient number" $\rightarrow$ "capture a sufficient number"

L. 8 Unclear terminology "significant training data"?

L. 9 Unclear terminology "threshold probabilities"

L. 24 " ... the estimated forecast errors meet the observed errors of the ensemble mean." This is confusing to me. You refer to the consistency of the spread of an ensemble and the ensemble mean forecast error in a perfect ensemble.

L. 26 "... high atmospheric variables ..." $\rightarrow$ "... atmospheric variables in higher vertical layers ..."

L. 30 "This calibration should be done in respect to maximize forecast sharpness" Apart from linguistic errors, it is exactly the opposite, statistical postprocessing maximizes sharpness on the condition of calibration.

L. 85 "..., since predictors also need to be detected that are not correlated to extreme events in order to ..." This sentence is confusing. You may refer here to the 'Forecaster's dilemma: Extreme events and forecast evaluation' by Lerch et al. in 2017.

L. 88 Ben Bouallégue et al. (2015) shows how to derive probabilistic forecasts for specific cost-loss scenarios.

L. 91 "... warning issuance" $\rightarrow$ "... warning insurance" ?

L. 135 "underdispersivity" $\rightarrow$ "underdispersion" ?

L. 171ff Not sure to understand this paragraph.

L. 195ff Not clear what and why you are doing this.

L. 259 The cost function is indeed the likelihood of the binomial distribution.

Eq. 2 It is unfortunate to use $p$ as the number of predictors, since $p$ is generally associated to probability. Later on this lead to some confusion (line 313).

Eq. 2 Clarify that $\hat{y}_p$ is a probability, and the observation $y$ is a binary variable taking values 0 and 1 (Bernoulli trials).

L. 263 "logarithm" $\rightarrow$ "logarithm of the likelihood"

L. 336 You say its effective, but you do not show anything in this respect. I may also be interesting to see, where Ensemble-MOS in not effective (e.g. what about fog).

L. 385 You may refer here to approaches to restore dependence (ensemble copula coupling, Shaake shuffle (e.g. Schefzik, 2016 and others).

---

## Author Comment (AC1) · 27 Apr 2020

**Author Comment to Anonymous Refereee #1**

See author comments in green colour

**General comments**

The article describes DWD's system to postprocess ensemble forecasts using the classical techniques linear and logistic regression with a focus on severe weather. While there is nothing new in the statistical methodology, the most interesting part lies in how the methods are applied to the data. The descriptions of the selection and design of predictors, how predictions are made at any location, how to deal with model changes, rare events etc. are all of interest to readers involved in operational forecasting. Possible weaknesses of the system could be discussed in more detail along with prospects of using more modern statistical methods. Please see below for more specific comments.

**Specific comments**

● line 63: Not sure I understand what the meaning of "... in order to avoid underestimation of forecast errors on longer time scales"
Sentence clarified, see introduction now. Idea is that MOS forecasts converge when the ensemble members are processed individually.

● line 106: As I understand, the combination is done after calibrating each model, but there seems to be no information about how this is done, apart from the citation of Reichert et al. (2015). Maybe a short paragraph on this interesting issue could be added to section 2.
Some information is added at the end of the introduction

● line 110. I find "Ensemble-MOS" a bit confusing here. Maybe it could be added that it is the name of DWD's system rather than referring to all "EMOS" methods.
Done

● line 135, Fig. 4d: Why do you use absolute errors instead of squared errors which is both more precise and commonly used?
the absolute errors show higher linear correlations, explanation clarified

● Fig. 4d: What is meant by "forecast of absolute error (m/s)"? Do you make predictions of the absolute errors directly or do you compute the absolute error after the gust forecasts are made?
4d left: this is basically the ensemble spread.of COSMO-DE-EPS
4d right: There are predictands for forecast errors, see Eq. 3, caption of Fig 4 (now 6) is improved

● line 140: Is the upscaled product an ensemble of 10x10x20 members or 20 members?
20 members, clarifed in text

● line 145: What kind of bias metric is this, given that the forecasts are probabilities exceeding 0.1 mm/hour?
If the foreast probabilities are measured in percent, their differences are percentage points (measuring differences in percent would be misleading)

● lines 160-165: Here results using Ensemble-MOS are presented before Ensemble-MOS is described in section 3. This is unfortunate and can easily cause misunderstandings, e.g. it is not obvious that Ensemble-MOS produce deterministic forecasts. Please consider moving this paragraph.
Manuscript has been restructured extensively

● line 167: As I understand, it is also used to make deterministic forecasts for continuous variables. Since this is a bit unusual, it should be stated explicitly somewhere, the sooner the better.
done

● lines 172-174: An alternative explanation: linear regression applied individually to an ensemble results in an ensemble of conditional means which is not the same as the conditional distribution of the predictand/observation given the predictors. In practice, the more uncertainty there is, the more will this adjusted ensemble underestimate the uncertainty.
Good point, explanation added

● line 193: Is it predictands or predictors? The sentence is unclear.
Sentence clarified (it is predictands)

● line 195: Would "binary predictands" be more precise than "probability predictands"?
Paragraph removed since obsolete anyway

● lines 195-200: Are then predictions from the linear regression interpreted as probabilities of exceeding the given threshold? Is there any statistical justification of the approach? Any references? How good is this approach compared to logistic regression? Maybe this paragraph should be expanded somewhat.
Paragraph removed since obsolete anyway

● line 202: replace predictor by predictand?
predictand, thanks

● line 237: I do not understand the sentence "Modeling this error ...". Do you mean that separate linear regression with the absolute error as predictand are made to predict the absolute error?
yes, sentence rephrased

● line 240: Any reference or better?
calculation added

● line 243: Do you mean the regression coefficients are interpolated separately?
yes, separately, but in the same linear way

● line 254: I would claim that logistic regression is no longer state of the art, at least in the research literature.
maybe, statement restricted to operational statistical models

● Equation 3: Maybe it could be added that conditional independence is assumed
Done

● lines 276-277: Not sure I understand this sentence. E.g. if there are 30 events in the training set, you would restrict the number of predictors to three?
Yes, sentence clarified

● line 299: These are deterministic predictions of gusts, or?
Sentence clarified. I would rather say "continuous" instead of "deterministic".
I consider the wind speed forecasts as means of statistical distributions including errors, compare Sec. about forecast consistency

● lines 300-301: Have you considered adding predictors describing the sites, e.g. elevation, exposure, land type etc.?
yes, see Table 1

● lines 307: please consider using "deterministic wind gust forecasts"
"continuous forecasts" added, see comment to line 299. Hope that is ok.

● line 311: have you investigated whether the globally fitted logistic regression make well-calibrated probabilities for each station, say, for the lower thresholds to get sufficient data?
Yes, see Fig. 7 in the new manuscript

● lines 312-326: these paragraphs need more accurate explanations. What is the relation between logistic regression and the logistic distribution? What is the fitted logistic distribution used for? What is the purpose?
abstract rephrased, hope it is clearer now

● line 400-402: what about raw ensemble probabilities for logistic regression?
Using the raw ensemble probablities as predictors could be an option. But the ensemble is underdispersive and information is lost. I suggest logistic regression to transfer from continuous variables to probabilities.
Do you suggest an EMOS approach (e.g. Gneiting?)

**Technical corrections**

● line 40: Is "landmarks" appropriate?
"locations" instead of "landmarks"

● line 57: Should it be postprocessing?
yes

● line 60: "coupula" should be "copula"
done

● line 62-63: maybe just "... tailored to postprocessing ensembles for extreme and rare events"?
done

● Figure 2: What is "MA EMOS"? Not easy to understand "Ensemble-MOS forecast of absolute error of Ensemble-MOS" either.
Caption rephrased. Is it clear now? Now it is Fig. 9

● line 202: each predictand(?), ...
"each predictand" is correct

● Equation 4: Parentheses are missing(?). Summation should be over both terms, I presume.
done

● line 299: stepwise linear regression(?)
Done

● figure 4: text in the plots is too small. What are the dashed lines?
dashed lines are mean +/- standard deviation as stated in the global caption of Fig. 6

● figure 6, caption: "is between"
done

**Author Comment to Anonymous Refereee #2**

See author comments in green colour

**General comments**

The manuscript describes the so-called Ensemble-MOS as operated at Deutscher Wetterdienst. The statistical methods applied in Ensemble-MOS are quite standard. Ensemble-MOS is based on univariate multiple linear and logistic regression. The interesting aspects of this manuscript is the operational implementation, which uses a large amount of predictors and predictants and a stepwise variable selection. The manuscript contains interesting aspects of how to apply postprocessing in an operational setting. Nevertheless, an extensive revision of the manuscript is necessary, especially with regard to structure and language. Further, several aspects should be examined more closely.

The manuscript has been revised.extensively

**Specific comments**

1. The general readability of the manuscript should be greatly improved. A clear separation in sections of (i) general aspects in postprocessing with literature, (ii) observations and forecasts, (iii) statistical methodology, (iv) DWD specific implementation and (v) results would largely increase the readability of the manuscript.
The structure has been revised accordingly.

2. A thorough revision of the English language is necessary. Language editing should also pay attention to the clarity in terminology.
The language has been improved, hope the manuscipt is better readable now.

3. Since the manuscript employs standard statistical methods, the description of the statistics may be shortened. On the other hand, the description of the nonstandard approach (why?) for wind gusts (section 3.3) is very confusing.
The description of the standard methods have been shortened as far as possible to keep the things readable that are nonstandard or at least interesting.
Section 3.3 is better motivated and clearified.

4. The results and particularly the figures should be discussed more closely. The order the figures are mentioned in the article must be consistent with the figure number.
The figures are now not anymore mentioned before they are discussed. I think this caused a lot of misunderstanding.

5. The title emphasizes on severe weather. The manuscript describes both, and severe weather is restricted to wind gusts. So I suggest investigate postprocessing for severe weather more closely. No results are given with respect to the warning issue. No scores are presented, and no comparison is provided of the performance of the not-postprocessed ensemble and the benefit obtained with postprocessing.
References to the warning issue are added and according results are mentioned.
Figs. 6 and 7 (in the revised manuscript) provide the requested comparisons.

6. No results a given with respect to the selected predictors, particularly for severe weather it might be interesting, which variables are most informative. Table 1 now presents a table of selected predicors for wind gusts. However, I am not convinced that this information is useful for physical interpretation.

7. More details could be provide on the effect of model changes. In line 354ff you discuss the inclusion of an indicator. Do you detect any sudden changes? More details may also be provided, how the change from COSMO-DE-EPS to COSMO-D2-EPS affects the postprocessing (you only mention small improvements in COSMO-D2-EPS in lines 363-365).
The change from COSMO-DE-EPS to COSMO-D2-EPS did not affect the postprocessing – surprisingly. No sudden changes were detected. Verifications show almost identical results with or without training with reforecasts of COSMO-D2-EPS.

**Minor comments and typos**

L. 2 "support warning decision management" □> "support the warning decision management"
Done

L. 8 "capture sufficient number" □> "capture a sufficient number"
Done

L. 8 Unclear terminology "significant training data"?
"significant training data" is training data that capture a sufficient number of cases with observed events, as required for robust statitical modelling, as stated in the text. Expression changed to "training data" to make it simpler.

L. 9 Unclear terminology "threshold probabilities"
"threshold probabilities" are probabilities that the strengths of events exceed given thresholds. Sentence is clarified.

L. 24 " ... the estimated forecast errors meet the observed errors of the ensemble mean." This is confusing to me. You refer to the consistency of the spread of an ensemble and the ensemble mean forecast error in a perfect ensemble.
yes, sentence rephrased

L. 26 "... high atmospheric variables ..." □> "... atmospheric variables in higher vertical layers ..."
done

L. 30 "This calibration should be done in respect to maximize forecast sharpness" Apart from linguistic errors, it is exactly the opposite, statistical postprocessing maximizes sharpness on the condition of calibration.
We agree on the content. I think it is ok to read "with respect to maximise ..." instead of "in respect to maximise...", does it?

L. 85 "..., since predictors also need to be detected that are not correlated to extreme events in order to ..." This sentence is confusing. You may refer here to the 'Forecaster's dilemma: Extreme events and forecast evaluation' by Lerch et al. in 2017. Sentence rephrased and further explanation added. Lerch et al. referred.

L. 88 Ben Bouallégue et al. (2015) shows how to derive probabilistic forecasts for specific cost-loss scenarios.
Ben Bouallegue et al. is referred now in the introduction

L. 91 "... warning issuance" ▢> "... warning insurance" ?
"...warning issuance" -> "automated warning guidance"

L. 135 "underdispersivity" ▢> "underdispersion" ?
done

L. 171ff Not sure to understand this paragraph.
Paragraph rephrased

L. 195ff Not clear what and why you are doing this.
Paragraph removed since obsolete

L. 259 The cost function is indeed the likelihood of the binomial distribution.
good point, adopted

Eq. 2 It is unfortunate to use p as the number of predictors, since p is generally associated to probability. Later on this lead to some confusion (line 313).
done, p -> k

Eq. 2 Clarify that ^yp is a probability, and the observation y is a binary variable taking values 0 and 1 (Bernoulli trials)
done

L. 263 "logarithm" ▢> "logarithm of the likelihood"
done

L. 336 You say its effective, but you do not show anything in this respect. I may also be interesting to see, where Ensemble-MOS in not effective (e.g. what about fog).
Statement restricted to wind gusts. Focus is on wind gusts where automatic warnings are most promising. I consider Ensemble-MOS as reference for more advanced sysstems.

L. 385 You may refer here to approaches to restore dependence (ensemble copula coupling, Shaake shuffle (e.g. Schefzik, 2016 and others).
cited in the introduction

---

## Author Response (AR2)

**Author Comments to Editors Decision**

See author comments in green colour

Comments to the Author:

The manuscript has been revised thoroughly according to the reviewers' comments. In the present form, this paper provides a comprehensible overview of post-processing routines run operationally at DWD, which is definitely relevant to the audience of this NPG special issue. Hence, I think the paper will be ready for publication subject to a very minor revision addressing the following issues:

- L50: EMOS does not necessary rely on Gaussian distributions (e.g. EMOS for precipitation based on shifted GEV or shifted Gamma distributions). So I would either call the method NGR (non-homogenous Gaussian regression) here, or rephrase the sentence in order emphasize that depending on the variable of interest EMOS methods relying on different parametric distribution families exist.

Thanks for the clarification. Method called NGR and sentence is rephrased appropriately.

- Equation (1): Why is a subscript k attached to predictand y. This notation is somewhat unusual and needs to be clarified.

The predictand is y.

\hat y_k is the estimation using a number of k predictors. The explicit labelling with k is used in the prescription of the stepwise regression in the paragraph after Eq. 7, lines 277ff-

Sentence is rephrased for clarification.

- L193: is this still the case (considering the developments in statistical post-processing over the last decade)?

Actually, I think it is.

Nevertheless, statement changed to „a classical approach".

- L213: What do you mean by variable here? NWP output?

Yes. „independent variables" removed since redundant here.

Some typos (list probably not exhaustive):

all corrected, thanks
- L48: forecast errors
- L63/64: use of calibrated ensembles
- L89: thunderstorms?
- L137: Too many observations
- L157: Bougeault et al. (2009); Swinbank et al. (2016)
- L479 : ranked probability score

Attached is an output of latexdiff to track the manuscript changes. Some minor corrections are included in addition. The References of Bougeault and Swinbank are correct in the manuscript, but fail to show up in the difference.

[revised manuscript text omitted]

---

## Author Response (AR3)

**Author Comments to Editors Decision 20 Jul 2020**

See author comments in green colour

All done. Some additional minor corrections have been made, see marked-up differences below.

- Line 13: Ensemble-MOS

- Line 46: to the costs of --> at the cost of

- Line 55: What do you mean by 'Many special systems'? Maybe something like 'Many different post-processing methods tailored to different variables exist, only...' would be easier to understand?

- Line 62/63: Difficult to understand. I would replace it by 'E.g., Schefzik et al. (2013); Schefzik and Möller (2018) use ensemble copula coupling (ECC) and Schaake shuffle-based approaches in order to generate post-processed forecast scenarios for temperature, precipitation and wind.'

- L105: I would write something like 'The results shown here focus on ...'

- L201 and 202: Do you really mean 'estimation'? I would call it an 'estimate'.
Done. I also replaced estimation with estimate at other places in the text

- L355: Maybe 'writing. Hence, the time series are subject to a number of model changes.' instead of 'writing, while a number of model changes have taken place.' would be easier to understand
- L414: 'TIGGE, see 2.3.' ◊ 'TIGGE, see Sec. 2.3.'

- Figure 10: Why are the consistency bars centered around the diagonal? Should the consistency bars not by more or less centered around the green or blue curve, respectively?

The consistency bars show the expected deviations from the diagonal due to sampling errors, under the assumption that the forecasts are perfectly reliable. In this way it can be decided if the deviations are still consistent with reliability. The reference Bröker and Smith (2006) is mentioned.
For my point of view this is more precise than showing significance bars, which define the range of the probability forecasts due to sampling errors (these significance bars are usually centered around the curves)

**Marked-up differences:**

[revised manuscript text omitted]

---

## Author Response (AR4)

**Author Comments to Editors Decision 11 Aug 2020**

Thank you again for reading the manuscript very thoroughly again and for your comments, especially for your helpful suggestions. I adopted all your comments and I think the manuskript has further improved.

I never had such an intense reviewing process before. Thank you for your help.

See my comments within the PDF-file. Following is the PDF with the topical differences of the manuscript.

Best regards,

Reinhold

[revised manuscript text omitted]